# Genetic Inactivation of Free Fatty Acid Receptor 3 Impedes Behavioral Deficits and Pathological Hallmarks in the APP_swe_ Alzheimer’s Disease Mouse Model

**DOI:** 10.3390/ijms23073533

**Published:** 2022-03-24

**Authors:** Marta Zamarbide, Eva Martinez-Pinilla, Francisco Gil-Bea, Masashi Yanagisawa, Rafael Franco, Alberto Perez-Mediavilla

**Affiliations:** 1Neuroscience Program, Center for Applied Medical Research (CIMA), University of Navarra, 31008 Pamplona, Spain; mzamarbide@unav.es (M.Z.); martinezpinillaeva@gmail.com (E.M.-P.); francisco.gilbea@biodonostia.org (F.G.-B.); 2Instituto de Investigación Sanitaria de Navarra (IDISNA), 31008 Pamplona, Spain; 3Department of Morphology and Cell Biology, Faculty of Medicine, University of Oviedo, 33006 Oviedo, Spain; 4Instituto de Neurociencias del Principado de Asturias (INEUROPA), 33003 Oviedo, Spain; 5Instituto de Investigación Sanitaria del Principado de Asturias (ISPA), 33011 Oviedo, Spain; 6International Institute for Integrative Sleep Medicine, University of Tsukuba, Tsukuba 305-8577, Japan; yanagisawa.masa.fu@u.tsukuba.ac.jp; 7Department of Biochemistry and Molecular Biomedicine, University of Barcelona, 08028 Barcelona, Spain; 8Network Center, Neurodegenerative Diseases, CiberNed, Spanish National Health Institute “Carlos III”, 28031 Madrid, Spain; 9Department of Biochemistry and Genetics, University of Navarra, 31008 Pamplona, Spain

**Keywords:** Alzheimer’s disease, neuroprotection, therapy, synaptic plasticity, learning and memory, amyloid

## Abstract

The free fatty acid FFA3 receptor (FFA3R) belongs to the superfamily of G-protein-coupled receptors (GPCRs). In the intestine and adipose tissue, it is involved in the regulation of energy metabolism, but its function in the brain is unknown. We aimed, first, to investigate the expression of the receptor in the hippocampus of Alzheimer disease (AD) patients at different stages of the disease and, second, to assess whether genetic inactivation of the Ffar3 gene could affect the phenotypic features of the APP_swe_ mouse model. The expression of transcripts for FFA receptors in postmortem human hippocampal samples and in the hippocampus of wild-type and transgenic mice was analyzed by RT-qPCR. We generated a double transgenic mouse, FFA3R^−/−^/APP_swe_, to perform cognition studies and to assess, by immunoblotting Aβ and tau pathologies and the differential expression of synaptic plasticity-related proteins. For the first time, the occurrence of the FFA3R in the human hippocampus and its overexpression, even in the first stages of AD, was demonstrated. Remarkably, FFA3R^−/−^/APP_swe_ mice do not have the characteristic memory impairment of 12-month-old APP_swe_ mice. Additionally, this newly generated transgenic line does not develop the most important Alzheimer’s disease (AD)-related features, such as amyloid beta (Aβ) brain accumulations and tau hyperphosphorylation. These findings are accompanied by increased levels of the insulin-degrading enzyme (IDE) and lower activity of the tau kinases GSK3β and Cdk5. We conclude that the brain FFA3R is involved in cognitive processes and that its inactivation prevents AD-like cognitive decline and pathological hallmarks.

## 1. Introduction

Alzheimer’s disease (AD) is a neurodegenerative disorder among the elderly, characterized by progressive deterioration of cognition, with a complex etiology involving multiple processes [1] Neuropathological features include senile plaques (SP) composed of aggregated amyloid β peptides (Aβ) derived from aberrant processing of the amyloid precursor protein (APP), neurofibrillary tangles (NFT) composed of hyperphosphorylated tau protein, and synaptic and neuronal loss [2]. In light of many recent studies, defects in lipid metabolism and consequent alterations in the membrane concentration of phospholipids may impact the development of pathophysiological features of AD [3,4,5,6,7,8]. In fact, the formation of SP and NFT correlates with decreased phospholipid biosynthesis and increased lipid degradation or peroxidation [9,10] that contribute to synaptic impairment and neuronal loss [11]. In addition, cumulative evidence indicates that inflammation occurs in vulnerable regions of the AD brain, contributing significantly to the pathogenesis [12,13,14]. Moreover, recent research regarding the major genetic risk factor has been conducted on the impact of APOE genotype on microflora speciation. Interestingly, metabolomic analysis detected significant differences in microbe-associated amino acids and short-chain fatty acids (SCFA) between APOE genotypes, indicating its association with specific gut microbiome profiles. These results suggest that further investigation related to the gut microbiome and fatty acids should be carried out to establish new targets able to mitigate the deleterious impact of the APOE4 allele on the cognitive decline associated with AD [15].

Free fatty acids (FFA) are part of phospholipids of neural cells and are important for the proper development and physiology of neurons [16,17]. Indeed, the brain lipid content is about 50% of its dry weight, confirming an essential role of lipids in the structure and functionality of the central nervous system (CNS) [18,19]. SCFA refer to carboxylic acids with aliphatic chains of less than six carbons, e.g., acetic, propionic, butyric or valeric acids. Long-chain fatty acids (LCFA) refer to carboxylic acids that are 13 to 22 carbons long, such as arachidonic (AA) and docosahexaenoic acids (DHA) often derived from food intake. Both SCFA and LCFA can be synthesized de novo or by salvage pathways. Interestingly, lipid homeostasis alteration in neurodegenerative diseases leads to altered lysosomal function and autophagy, which in turn results in impaired processing of APP [20,21,22,23].

FFA regulatory effects are mediated by the receptors that belong to the superfamily of G protein-coupled receptors (GPCRs). GPCRs constitute a large family of seven transmembrane domain membrane proteins that are activated by a huge variety of molecules. They play a key role in mammalian homeostasis. GPCR activation may engage a variety of signaling molecules and activate (or deactivate) a diversity of signaling pathways [24,25]. Five FFA receptors (FFARs) have been identified: FFA1R (or GPR40) and FFA4R (or GPR120), which are selectively stimulated by LCFA, and FFA2R (or GPR43), FFA3R (or GPR41), and GPR42, which are selectively stimulated by SCFA. Moreover, there is an unconfirmed FFAR candidate, GPR84, which would be activated by medium-chain fatty acids [26,27,28,29,30]. Whereas not much is known about the events mediated by GPR42, all other FFARs couple to G_q_, with the exception of the FFA3R, which couples to G_i_/G_o_ [28]. FFA3R also regulates the activation of the mitogen-activated protein kinase (MAPK) pathway [31].

These receptors have been studied in depth in peripheral tissues, especially in adipose tissues and in the pancreas ([31,32,33,34] for review). FFARs are involved, among others, in adipocyte differentiation [35], insulin release by the pancreas [36], and growth hormone release by the pituitary gland [37]. Their expression and function have been less investigated in the central nervous system (CNS), although the participation of FFARs in some neuropathies has been suggested [4,38,39].

Concerning FFA3R, the gene expression was analyzed by the GTEx Project in multiple reference human tissues by high-throughput RNAseq screening. The FFA3R expression pattern presents higher levels in adipose tissue, breast, colon, spleen, and digestive tract, while neural expression was lower. In more detail, within the CNS, the highest levels were found in the spinal cord, followed by substantia nigra, hypothalamus, caudate, and hippocampus [4,40].

Most of the studies regarding this receptor have been performed in the peripheral nervous system, demonstrating its expression on postganglionic sympathetic and sensory neurons in both the autonomic and somatic nervous system [41]. Moreover, it has been established that the FFA3R mediates the effects of SCFAs in sympathetic nervous system activity, thus contributing to the control of body energy expenditure and metabolic homeostasis [42].

As SCFAs are the main metabolites produced by the microbiota, they have a key role in the gut–brain crosstalk [43]. In line with this, recent studies focus on the study of this receptor in the gut–brain axis, whose recognized mechanisms are the modification of autonomic/sensorimotor connections, immune activation, and regulation of the neuroendocrine pathway [44]. There is growing evidence of the role of the microbiota in the regulation of physiology and behavior in metabolic, inflammatory, and neurodegenerative disorders [45]. In fact, amelioration of motor deficits and reduction in dopaminergic neuronal loss in a mouse model of Parkinson’s disease (PD) may be achieved via regulating FFA3R expression in the enteric nervous system by the bacteria that inhabit the gut [46].

Since lipids are important for brain structure and function, and since ketone bodies, which may activate the FFA3R, are an important energy source for neurons, we hypothesized that the receptor could be a target for AD. Accordingly, we focused on the potential expression of the FFA3R in the hippocampus and on assessing what would be the consequence of a lack of the receptor in a transgenic mice model. Hence, the aim of this work was to investigate the expression of the receptor in the hippocampus of AD patients and, second, to assess whether genetic inactivation of the Ffar3 gene could affect the phenotypic features of one AD mice model. For this purpose, a new transgenic line was developed by crossing the Tg2576 line (APP_swe_), which overexpresses the human APP carrying the Swedish mutation, with FFA3R knockout (KO) (FFA3R^−/−^) mice. 

## 2. Results

### 2.1. Expression of Transcripts for FFA Receptors in Postmortem Human Hippocampal Samples

The study of FFAR expression was performed in human hippocampi of postmortem non-demented (ND) individuals and individuals with AD. By RT-qPCR, we found both that transcripts for FFA1R, FFA3R, and FFA2R are expressed in the human hippocampus and that their expression is altered in AD. Considering Braak and Braak stages [47], there was a decrease in FFA1R levels in the early stages of AD, compared with ND controls (*p* < 0.001), and an increase with the course of the pathology. The increase, however, never reached the amount found in ND individuals (Figure 1a). FFA2R transcript levels were significantly decreased (50–60%) in AD patients in relation to ND controls (*p* < 0.001), without significant differences through the stages of the pathology (Figure 1b). Remarkably, FFA3R transcripts increased by circa 5-fold at the early stages of the disease (*p* < 0.001). The level was still increased in stage V, although it went down to the control at later stages (*p* < 0.01) (Figure 1c).

### 2.2. Expression of Transcripts for FFA Receptors in the Hippocampus of WT and Transgenic Mice

Hippocampal expression of FFAR in mice was assessed by RT-qPCR. We found a significant increase in FFA1R transcript expression levels in FFA3R^−/−^ mice (*p* < 0.001 vs. WT mice) and also in FFA3R^−/−^/APP_swe_ compared with APP_swe_ mice (*p* < 0.05). By contrast, a decrease in FFA1R mRNA level was found in samples from APP_swe_ mice (*p* < 0.01 vs. WT mice) (Figure 2a). Meanwhile, the FFA2R transcript level was significantly downregulated in APP_swe_ (*p* < 0.01), FFA3R^−/−^ (*p* < 0.05), and FFA3R^−/−^/APP_swe_ (*p* < 0.01) compared with WT mice (Figure 2b). Finally, FFA3R transcripts were overexpressed in the hippocampus of APP_swe_ mice (*p* < 0.001 vs. WT mice) (Figure 2c). 

### 2.3. Genetic Inactivation of FFA3R Reverses the Cognitive Impairment of APP_swe_ Mice

To assess spatial reference learning and memory function, groups of 12-month-old FFA3R^−/−^, FFA3R^−/−^/APP_swe_, and APP_swe_ mice and the corresponding age-matched control littermates (10–12 mice per group) were tested in the MWM. No statistically significant differences in escape latency among groups were found during the visible platform training (Figure 3a). In the spatial reference training (invisible platform), we found escape latencies significantly shorter in WT, FFA3R^−/−^, and FFA3R^−/−^/APP_swe_ than in APP_swe_ mice (*p* < 0.01) (Figure 3b).

The measurement of memory retention was performed in the pool without the platform. On day 4, no significant inter-group differences were found. On day 7, the APP_swe_ mice exhibited a significantly lower proportion of time spent in the target quadrant compared with WT, FFA3R^−/−^, or FFA3R^−/−^/APP_swe_ mice (APP_swe_ vs. WT, *p* < 0.01; APP_swe_ vs. FFA3R^−/−^, *p* < 0.001; APP_swe_ vs. FFA3R^−/−^/APP_swe_, *p* < 0.05). On day 9, the APP_swe_ mice exhibited again a significantly lower proportion of time spent in the target quadrant compared with WT, FFA3R^−/−^, and FFA3R^−/−^/APP_swe_ mice (*p* < 0.05) (Figure 3c). The swim speed did not differ significantly between groups, and the distance data exhibited the same pattern as the escape latency (data not shown). These results demonstrate that FFA3R^−/−^/APP_swe_ mice show a better cognition status as assessed by the MWM test.

### 2.4. Aβ Pathology Is Reversed in the Brain of FFA3R^−/−^/APP_swe_ Mice

Aβ burden is one of the most relevant AD-like features in the APP_swe_ line. The effect of Ffa3R knockdown on Aβ production and fibrillar Aβ depositions was determined. ELISA measurement in cortical homogenates showed barely detectable levels of Aβ_42_ in 12-month-old FFA3R^−/−^/APP_swe_ mice, as opposed to what happened in age-matched APP_swe_ mice, which had high levels of this polypeptide (Figure 4a). No Aβ_42_ was detected in non-transgenic littermates and FFA3R^−/−^ mice (data not shown). 

We analyzed the APP-derived CTFs processing in these animals by Western blot. As it is shown in Figure 4b, there were no significant differences in the levels of the CTFs (C83 and C99) between FFA3R^−/−^ vs. WT and FFA3R^−/−^/APP_swe_ vs. APP_swe_ mice.

Interestingly, immunohistochemical analysis of hippocampal and frontal cortex sections using 6E10 antibody demonstrated that FFA3R^−/−^/APP_swe_ mice were completely free of Aβ deposits, which were abundant in the brain of the APP_swe_ mouse (Figure 4c). Concerning this, we found that the levels of insulin degrading enzyme (IDE), which reduces the accumulation of APP-derived toxic peptides, were increased in the hippocampus of FFA3R^−/−^ and FFA3R^−/−^/APP_swe_ mice (FFA3R^−/−^ vs. WT, *p* < 0.01; FFA3R^−/−^/APP_swe_ vs. WT, *p* < 0.001; FFA3R^−/−^/APP_swe_ vs. APP_swe_, *p* < 0.01). Therefore, the Aβ reduction, both soluble and fibrillar, could be a consequence of higher clearance mediated by Aβ degrading enzymes. Whereas IDE was increased, no significant inter-group differences in the expression of neprilysin transcripts were detected (Figure 4d,e).

### 2.5. Hippocampus of FFA3R^−/−^/APP_swe_ Mice Show a Decrease in Tau Pathology

Hyperphosphorylation of tau protein is another of the main hallmarks of AD, so we explored the possible neuropathological correlation of phosphorylated tau levels and the memory improvement in FFA3R^−/−^/APP_swe_ mice. The 12-month-old FFA3R^−/−^/APP_swe_ mice displayed significantly reduced levels of phosphorylated tau, at the epitopes recognized by the AT8 antibody (Ser^202^/Thr^205^), compared with the APP_swe_ transgenic mice. No significant change in phosphorylated tau levels was found in the FFA3R^−/−^ mice compared with non-transgenic controls (Figure 5a).

We then looked for changes in kinase activity accounting for the reduction in tau phosphorylation. The results show an increase in the tau kinase GSK3β active form, phosphorylated at Tyr^216^, in APP_swe_ mice compared with WT mice (*p* < 0.001) (Figure 5b). FFA3R^−/−^/APP_swe_ mice showed significantly lower levels of active GSK3β as compared with APP_swe_ mice (*p* < 0.001) and similar to WT mice. In addition, the inactive form of GSK3β, phosphorylated at Ser^9^, was increased in FFA3R^−/−^/APP_swe_ vs. APP_swe_ (*p* < 0.001) (Figure 5c). 

It has been demonstrated that the kinase Cdk5, which also phosphorylates tau, is activated by p35 protein. Calpain-mediated cleavage of p35 to p25 (a 208-residue carboxy-terminal fragment of p35) p25 causes prolonged activation and mislocalization of Cdk5 and hyperphosphorylation of substrates such as tau. The analysis of the p25/p35 ratio in APP_swe_ mice showed a significant increase (1.63 ± 0.07 (*p* < 0.01)), whereas FFAR3 ^−/−^/APP_swe_ (0.75 ± 0.10) remained with similar ratios to FFA3R^−/−^ and WT (Figure 5d). 

### 2.6. FFA3R^−/−^/APP_swe_ Mice Show an Increase in Mature BDNF and the Expression of Synaptic Plasticity-Related Proteins

Memory consolidation is a process for stabilizing short-term memory, generating long-term memory by the expression of several immediate early genes (IEGs) [48]. To assess the biochemical substrate for memory recovery, we analyzed the expression of several IEG products. In order to induce the expression of genes related to memory consolidation, animals were given fear conditioning training and sacrificed 2 h later (Figure 6a). Then, the levels of the synaptic activity-dependent proteins (c-Fos and Arc) and the phosphorylation of the CREB transcription factor were determined by Western blot using hippocampal samples from these animals. Compared with data in samples from WT mice, the hippocampus of APP_swe_ animals showed a reduction of 52 ± 7% (*p* < 0.001), 68 ± 3% (*p* < 0.05), and 66 ± 5% (*p* < 0.01) in c-Fos, Arc, and pCREB levels, respectively. However, we did not find differences in the amount of these proteins when comparing samples from WT, FFA3R^−/−^, and FFA3R^−/−^/APP_swe_ mice (Figure 6b–d).

The brain-derived neurotrophic factor (BDNF) is involved in neural synaptic development. Compared with data in samples from WT mice, the hippocampus of APP_swe_ animals showed a decrease in mBDNF/proBDNF ratio (*p* < 0.05) due to a significant increase in proBDNF levels (*p* < 0.01). Although not significant, the mBDNF/proBDNF ratio in samples from FFA3R^−/−^/APP_swe_ was in between from those found in WT and APP_swe_ mice (Figure 7).

To better understand the basis of good performance in the MWM test displayed by 12-month-old FFA3R^−/−^/APP_swe_ mice, we also wanted to address synaptic plasticity in the cortex as it is also involved in AD pathology and the consolidation of new memory circuits. First, the ionotropic AMPA receptors were found increased in both FFA3R^−/−^ and FFA3R^−/−^/APP_swe_ mice. In fact, the results show a statistically significant increase in the phosphorylation of the AMPA subunit GluR1 in FFA3R^−/−^ and FFA3R^−/−^/APP_swe_ mice compared with WT and APP_swe_ mice, respectively (*p* < 0.001) (Figure 8a). Meanwhile, the analysis of GluR2 and GluR3 AMPA subunits exhibited an increase in their expression levels in FFA3R^−/−^/APP_swe_ vs APP_swe_ mice and WT mice (*p* < 0.05) (Figure 8b).

Cytoskeletal protein PSD95, a member of the post-synaptic density, is decreased in APP_swe_ mice (*p* < 0.05 vs. WT mice), and its levels are recovered in the FFA3R^−/−^/APP_swe_ mice (*p* < 0.001 vs. APP_swe_ mice) and in FFA3R^−/−^ compared with WT mice (*p* < 0.001) (Figure 8c). MAP-2 shows similar cortical levels in the WT, FFA3R^−/−^, and FFA3R^−/−^/APP_swe_ mice. Conversely, APP_swe_ mice have decreased levels of MAP-2 compared with WT mice (*p* < 0.05) (Figure 8d).

Finally, the levels of the phosphorylated form of CaMK II are preserved in FFA3R^−/−^ and FFA3R^−/−^/APP_swe_ mice, whereas they are decreased in the APP_swe_ mice (*p* < 0.01 vs. WT mice) (Figure 8e). Comparing the transgenic animals, the expression was higher in FFA3R^−/−^/APP_swe_ than in APP_swe_ mice.

## 3. Discussion

Over the past decades, researchers have made remarkable progress in understanding the molecular events that lead to the characteristic progressive decline in cognition and memory in AD [49,50]. Despite these efforts, current therapies for AD can only temporarily improve symptoms of memory loss without delaying the course of the disease—i.e., the death of neurons and the decline in higher cognitive functions [51]. Growing evidence demonstrates that defects in lipid metabolism and, consequently, alterations in vesicular dynamics and synaptic plasticity are related to the pathogenesis of AD [4,5]. Recently, it has been demonstrated that the plasma lipidome is dysregulated in AD. Specifically, AD patients show several alterations in levels of different lipid classes: sphingomyelins, cholesterol esters, phosphatidylcholines, phosphatidylethanolamines, phosphatidylinositols, and triglycerides, but no differential results were obtained in relation to SCFA lipids. SCFAs such as propionate or valerate, whose deficiency has been associated with AD, can influence plasma membrane organization and activity through the FFA3R [31,52]. In fact, FFA3R is strongly expressed in the gut, where it has been studied extensively in intestinal motility and inflammation [53,54]. However, its expression and the role it may play in the CNS has not yet been adequately addressed. Although there is some evidence of their presence in the nervous system [4], the lack of studies in relation to these receptors is probably due to the difficulty of their characterization at the protein level due to the absence of specific antibodies.

In this study, we demonstrated for the first time that FFARs are expressed in the human hippocampus of both control and AD patients. In addition, data on mRNA transcription levels in postmortem hippocampal tissue from AD patients showed FFA3R overexpression in the early stages of the disease whereas FFA1R and FFA2R expression appeared downregulated. We then reasoned that a transgenic mouse obtained by crossing an AD mouse model with FFA3R KO mice would provide insights into the potential of FFARs as therapeutic targets in the disease. We chose the APP_swe_ transgenic mouse, which is considered an early AD model [55], because it has a hippocampal pattern of FFA3R expression that mimics that found in patients at the early stages of the disease. As the most notable behavioral characteristic in AD models is memory impairment, we noted using the MWM test that the lack of FFA3R per se does not interfere with spatial memory. However, the main result of the study is that the FFA3R genetic defect (FFA3R^−/−^/APP_swe_ mice), which is overexpressed in the hippocampus of patients, completely reverses the characteristic cognitive deficits of 12-month-old APP_swe_ mice [48,55,56,57]. Differential expression in patients vs controls and this specific finding in double transgenic mice leads to the conclusion that blocking FFA3R function prevents memory loss in the context of AD.

Memory consolidation involves the expression of IEGs, which play a critical role in the transformation of neural events for the acquisition into long-term memories, mainly through cortico-hippocampal circuits [58,59]. IEG products such as pCREB, Arc, and c-Fos are also reduced in the cerebral cortex and the hippocampus of another AD mice model, the APP/PS1 transgenic mice [60,61]. Aβ pathology also impacts on reducing the expression of functional Arc [62]. In this regard, our work shows that the lack of FFA3R restores the aberrant expression of IEGs in mice that overexpress the APP_swe_, making them similar to the levels found in WT mice. There is little information on the consequences of the activation of FFARs expressed in the CNS. It should be noted that, at the mechanistic level, the activation of FFA3R may induce the phosphorylation of CREB. The mechanism may be independent of the alpha subunit of the canonical heteromeric G protein of the receptor, G_i_, because engagement of this protein would lead to a reduction in the levels of cAMP. We favor the hypothesis of involvement of beta-gamma subunits in linking receptor activation to CREB phosphorylation and IEG expression. A previous study shows that activation of FFAR1 induces the phosphorylation of CREB and of extracellular-regulated kinase (ERK) 1/2 in primary hippocampal neurons and in a human neuroblastoma cell line [25]. CREB signaling is known to regulate the expression of genes that promote synaptic and neuronal plasticity, including the gene for brain-derived neurotrophic factor (BDNF).

Synaptic activity plays a fundamental role in the consolidation of new memory circuits, and different enzymes, receptors, and cytoskeletal proteins, whose levels are decreased in patients with AD, are involved [63]. APP_swe_ mice show a reduction in the cytoskeletal proteins of dendritic spines such as PSD95 and MAP-2, probably related to the accumulation of Aβ [64]. Due to their role in memory consolidation and long-term improvement, these characteristics are also present in AD patients [65]. Interestingly, we have found a similar expression of MAP-2 in FFA3R KO mice compared with WT mice, and higher levels of PSD95 in FFA3R^−/−^ and FFA3R^−/−^/APP_swe_ compared with WT mice. These findings might explain the restoration of cognitive status observed in FFA3R^−/−^/APP_swe_ mice. Moreover, we studied the processing of BDNF, a neurotrophic factor that plays a key role in neuronal survival and, among others, in synaptic function and cognition [66]. ProBDNF is subsequently cleaved to form a mature protein, mBDNF [67,68], which has been associated with AD [69]. O’Bryant et al. (2009) have documented a decrease in BDNF mRNA levels in postmortem brain samples of patients diagnosed with AD and mild cognitive impairment [70]. It has also been suggested that serum BDNF levels are altered in AD [71,72]. Consistent with the main conclusion of this article, the decrease in the mBDNF/proBDNF ratio in APP_swe_ mice (compared with WT) was restored in the FFA3R^−/−^/APP_swe_ ones.

The significant decrease in sAβ content and the absence of SP in the hippocampus and cortex of the FFA3R^−/−^/APP_swe_ mice were consistent with the correlation of sAβ levels and the synaptic alterations and cognitive dysfunction exhibited by the transgenic AD mice [73,74]. The absence of SP in the FFA3R^−/−^/APP_swe_ mice could be due to canonical APP proteolysis, but the alternative hypothesis—that is, an increase in Aβ clearance—seems more logical [75,76,77]. In fact, we did not observe any deficits in APP processing since the levels of C83 and C99 were identical to those of APP_swe_ mice. Interestingly, FFA3R^−/−^/APP_swe_ mice show a significant increase in the expression levels of the IDE enzyme, while no differences were found in the analysis of the expression levels of NEP. These data support the hypothesis that IDE promotes an increase in Aβ clearance that reduces the accumulation of this harmful peptide in the brain. Cognitive impairment in transgenic AD models is also accompanied by accumulations of phosphorylated tau, which are composed of aberrant forms of phosphorylated tau [78,79,80]; indeed, higher activities of GSK3β and Cdk5 kinases result in accumulation of hyperphosphorylated tau in 12-month-old APP_swe_ mice [81,82,83]. The increase in the inactive pGSK3β-Ser^9^ form in the FFA3R^−/−^/APP_swe_ mice, along with a parallel decrease in the active pGSK3β-Tyr^216^ form, may explain the significant decrease in aberrant tau phosphorylation.

## 4. Materials and Methods

### 4.1. Human Brain Samples 

Human brain samples (specifically, the hippocampal region) came from the Navarra Health Service/Osasunbidea’s Research Biobank and came from patients who met the criteria for the diagnosis of AD-like pathology with Braak stages from III to VI and symptoms of dementia, and from non-demented individuals with no AD pathology (Table 1). All procedures were carried out in accordance with the Ethics Committee of the University of Navarra and, for all subjects, written informed consent for whole body autopsy, and for the removal of all organs for diagnostic and research purposes, was obtained from their next of kin. Brain samples were stored at −80 °C until processing.

### 4.2. Animals and Behavior Test

In this study, we used groups of 12-month-old mice (n = 44; n = 24 females and n = 20 males) of four genotypes: APP_swe_, FFA3R^−/−^/APP_swe_, FFA3R^−/−^ mice, and WT littermates. APP_swe_ mice expressing the human 695-aa isoform of the APP containing the Swedish double mutation (APP_swe_) ((APP695)Lys670→Asn, Met671→Leu) driven by a hamster prion promoter were used as a mouse model of AD [48]. FFA3R^−/−^ and FFA3R^−/−^/APP_swe_ mice were generated in our laboratories by breeding FFA3R^−/−^ mice with APP_swe_ mice [84]. The animals were maintained in positive pressure-ventilated racks at 25 ± 1°C with a 12 h light/dark cycle, fed ad libitum with a standard rodent pellet diet (Global Diet 2014; Harlan), and allowed free access to filtered and UV-irradiated water. All animal care and experimental procedures were in accordance with European and Spanish regulations (86/609/CEE; RD1201/2005) and were approved by the Ethical Committee of the University of Navarra (no. 018/05). Behavioral studies were carried out during light time (from 9:00 a.m. to 2:00 p.m.). Details about the genotyping of the different mice strains can be found in Appendix A.

### 4.3. Morris Water Maze Test (MWM)

Groups of 12-month-old APPswe, FFA3R^−/−^/APP_swe_, and FFA3R^−/−^ mice and WT littermates, underwent spatial memory analysis using the Morris water maze (MWM) test. MWM allows analysis of hippocampus-dependent learning as described previously [85]. The day before the behavioral test, animals were housed in the ad hoc room for acclimatization and maintained in the room until test completion. The water maze was a circular tank (diameter 1.2 m) filled with water at 20 °C and made opaque by the addition of non-toxic white paint. Mice underwent visible-platform training for three consecutive days (6 trials/day) and were allowed to swim to a raised platform (diameter 10 cm) located above the water in the same position over the trials. No distal visible cues were present during this phase. Hidden-platform training was conducted over 8 consecutive days (4 trials/day). Mice had 60 s to find a hidden platform submerged 1 cm beneath the surface of the water and invisible to the mice while swimming. Several large visual cues were placed in the room to guide the mice to the hidden platform. In both visible- and hidden-platform versions, mice were placed pseudo-randomly in selected locations, facing towards the wall of the pool to eliminate the potentially confounding contribution of extramaze spatial cues. Mice that failed to reach the platform were guided onto it. All of the animals were allowed to rest on the platform for 20 s and were then removed from the platform and returned to their home cage. At the beginning of the 4th, 7th, and 9th day of the task, a probe trial, in which the platform was removed from the pool, was conducted, and the mice were permitted to search for the platform for 60 s. All trials were monitored by a camera above the center of the pool connected to a SMART-LD program (Panlab S.L., Barcelona, Spain) for subsequent analysis of escape latencies, swimming speed, path length, and percentage of time spent in each quadrant of the pool during the probe trials. Mice that were unable to reach the visible platform and those exhibiting abnormal swimming patterns or persistent floating were excluded from data analyses. All experimental procedures were performed blind to groups.

### 4.4. Fear Conditioning-Based Memory Induction Protocol 

To induce the immediately early genes, all the animals were placed under a cognitive stimulus based on a fear conditioning test as described previously [86]. Details can be found in Appendix A. Two hours after the fear conditioning test, the animals were killed, and the brains were removed for biochemical studies.

### 4.5. Brain Tissue Processing for Immunohistochemistry

Under xylazine/ketamine anesthesia, animals were perfused transcardially with saline and 4% paraformaldehyde (PFA) in phosphate buffer (PB). A 1 h post-fixation step with 4% PFA was carried out before the overnight cryoprotection step in 30% sucrose solution in PB at 4 °C. Microtome sections (30 µm thick) were cut coronally and stored in cryopreservation solution at −20 °C until processed. Immunohistochemistry was performed in nine free floating tissue sections comprising the hippocampal formation of three animals per group. Details about immunohistochemistry protocol can be found in Appendix A. 

### 4.6. Protein Extracts

Mice were killed by cervical dislocation and hippocampi were quickly dissected from the brains. Total tissue homogenates were obtained by homogenizing the hippocampus in a cold lysis buffer with protease inhibitors (0.2 M NaCl, 0.1 M HEPES, 10% glycerol, 200 mM NaF, 2 mM Na_4_P_2_O_7_, 5 mM EDTA, 1 mM EGTA, 2 mM DTT, 0.5 mM PMSF, 1 mM Na_3_VO_4_, 1 mM benzamidine, 10 µg·mL^−1^ leupeptin, 400 U·mL^−1^ aprotinin). After that, the homogenates were centrifuged at 14,000× *g* for 20 min at 4 °C, and the supernatant was aliquoted and stored at −80 °C. 

To obtain the membrane-enriched protein fraction (P2 membrane proteins), a previously described method [87] was used. The hippocampi were homogenized in ice-cold Tris-EDTA buffer (10 mM Tris-HCl and 5 mM EDTA, pH 7.4) containing 320 mM sucrose and the protease and phosphatase inhibitors previously described. The tissue homogenate was centrifuged at 700× *g* for 10 min. The collected supernatant was centrifuged again at 37,000× *g* for 40 min at 4 °C. Finally, the pellet (P2) was resuspended in 10 mM Tris-HCl buffer (pH 7.4), containing the enzyme inhibitor mixture described above. In both cases, protein concentration was determined (Bradford assay, Bio-Rad, Hercules, CA, USA) and aliquots were stored at −80 °C until use. For Western blot analysis, aliquots of the P2 membrane fraction were solubilized in denaturing conditions by adding 0.1 volumes of 20% SDS and 50% β-mercaptoethanol. The samples were incubated for 5 min at 100 °C and diluted 1:20 in 50 mM Tris-HCl (pH 9)/0.1% Triton X-100. After a centrifugation step at 37,000× *g* for 10 min at 4 °C, the supernatant was stored at −80 °C.

For APP-derived carboxy-terminal fragments (CTFs) determination, the cortex was homogenized in a buffer containing SDS 2%, Tris-HCl (10 mM, pH 7.4), protease inhibitors (Complete Protease Inhibitor Cocktail, Roche), and phosphatase inhibitors (0.1 mM Na_3_VO_4_, 1 mM NaF). The homogenates were sonicated for 2 min and centrifuged at 100,000× *g* for 1 h. Aliquots of the supernatant were frozen at −80 °C, and protein concentration was determined by the Bradford method. 

### 4.7. Determination of Aβ Levels

For the analysis of soluble Aβ_42_ burden, the same protein extraction that was used for APP-derived CTFs determination was followed. Soluble Aβ_42_ levels were determined by using a sensitive sandwich ELISA kit from Biosource (Camarillo, CA, USA) following the manufacturer’s instructions.

### 4.8. Immunoblotting

Protein samples were mixed with 6× Laemmli sample buffer resolved onto SDS-polyacrylamide gels [53] and transferred to nitrocellulose membrane. For the analysis of APP-derived CTFs, protein extracts were separated in a Criterion^TM^ precast Bis-Tris 4–12% gradient precast gel (Bio-Rad, Hercules, CA, USA) and transferred to nitrocellulose membranes. Details and a list with primary and secondary antibodies used can be found in Appendix A.

### 4.9. RNA Extraction and Quantitative Reverse Transcription PCR (RT-qPCR)

Total RNA was extracted from the brain hippocampus by Chomczynski and Sachi’s method [54] with the TRI reagent (Sigma-Aldrich, St. Louis, MO, USA). Details of RNA extraction and RT-qPCR as well as a table with the primer sequences for quantitative PCR can be found in Appendix A. 

### 4.10. Data Analysis and Statistical Procedures

The data were analyzed with SPSS for Windows version 15.0 (SPSS, Chicago, IL, USA) and, unless otherwise indicated, the data are expressed as the means ± standard error of the mean (S.E.M.). The normal distribution of data was checked by the Shapiro–Wilks test. In the MWM, latencies to find the platform were examined by two-way repeated measures ANOVA test (genotype × trial) to compare the cognitive status in WT mice, APP_swe_, FFA3R^−/−^ and FFA3R^−/−^/APP_swe_. Likewise, the spatial memory and the biochemical data were examined also by a two-way ANOVA test (Treatment × Trial) followed by post hoc Tukey’s analysis. When the interaction between factors was significant, single effects were analyzed by one-way ANOVA followed by post hoc Tukey’s test. When no significant interaction between factors was found, the main effects were analyzed. Student’s *t*-test was used when two groups were compared.

## 5. Conclusions

The data obtained in the present study indicate that FFA3R is expressed in the healthy human hippocampus and that a genetic deletion of the receptor results in the dissipation of the molecular and behavioral abnormalities shown by the APP_swe_ animal model. We hypothesize that the lack of FFA3R probably produces a different profile of FFAR expression that stimulates the clearance of soluble Aβ content by the activation of different signaling pathways, promoting the removal of senile plaques in the hippocampus and cortex by the increased expression of IDE. Consequently, FFA3R emerges as a therapeutic target for combatting AD. Very relevant is the information on the chemical structures that differentially affect FFA3R-mediated signaling. Inverse agonists, antagonists, and negative allosteric modulators are now available [34,40,88,89,90,91,92] and could easily be tested for safety and efficacy in animal models of disease. Last but not least, future work should address the role of FFA3R in brain development, neurogenesis, and neurophysiological functions. 

## Figures and Tables

**Figure 1 ijms-23-03533-f001:**
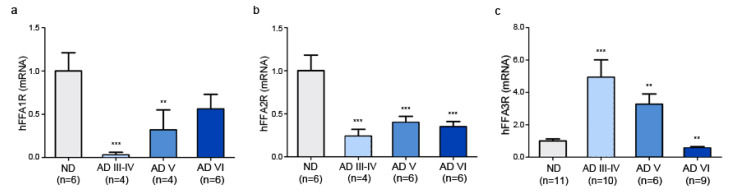
FFARs in postmortem hippocampus tissue from AD patients. mRNA levels of FFAR in hippocampal sections of AD patients and the corresponding ND controls analyzed by RT-qPCR. Bars represent the receptor mRNA expression normalized to that of the corresponding 36B4 internal control. Values are expressed as mean ± SEM. (**a**) Expression levels of FFA1R in ND and patients of different stages of AD. (**b**) Expression levels of FFA2R in ND and patients of different stages of AD. (**c**) Expression levels of FFA3R in ND and patients of different stages of AD. ** *p* < 0.01, *** *p* < 0.001 vs. ND.

**Figure 2 ijms-23-03533-f002:**
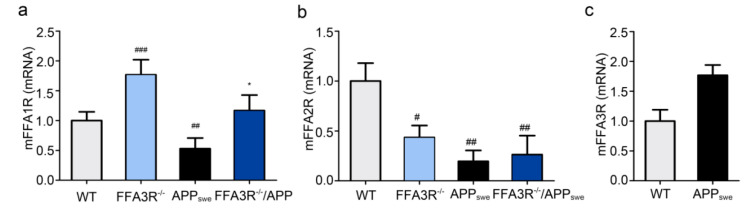
Comparative expression of FFARs in hippocampus of WT and transgenic mice. mRNA levels of FFAR in hippocampus of WT, FFA3R^−/−^, APP_swe_, and FFA3R^−/−^/APP_swe_ mice analyzed by RT-qPCR. Bars represent the receptor mRNA expression normalized to that of the corresponding 36B4 internal control. Values are expressed as mean ± SEM (n = 6) normalized to the ratio of WT mice. (**a**) Expression levels of FFA1R. (**b**) Expression levels of FFA2R. (**c**) Expression levels of FFA3R. # *p* < 0.05, ## *p* < 0.01, ### *p* < 0.001 vs. WT. * *p* < 0.05 vs. APP_swe_.

**Figure 3 ijms-23-03533-f003:**
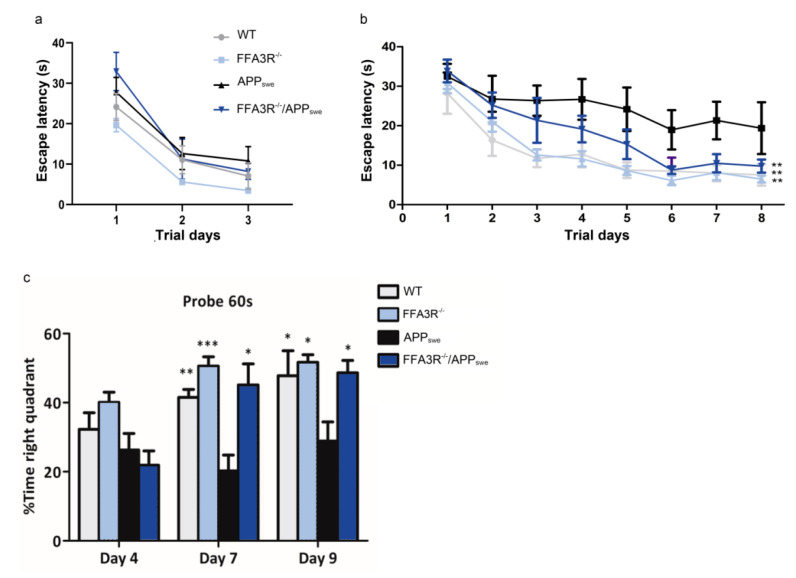
FFA3R^−/−^/APP_swe_ mice do not show the cognitive deficit of 12-month-old APP_swe_ mice. Escape latency times of the visible (**a**) and hidden platform (**b**) in the MWM test for the WT, FFA3R^−/−^, APP_swe_, and FFA3R^−/−^/APP_swe_ mice in the different trial days. (**c**) Percentage of time spent searching for the target quadrant of the probe test for the WT, FFA3R^−/−^, APP_swe_, and FFA3R^−/−^/APP_swe_ on day 4, 7, and 9. Results are expressed as mean ± SEM (n = 10–12 in each group). * *p* < 0.05, ** *p* < 0.01, *** *p* < 0.001 vs. APP_swe_.

**Figure 4 ijms-23-03533-f004:**
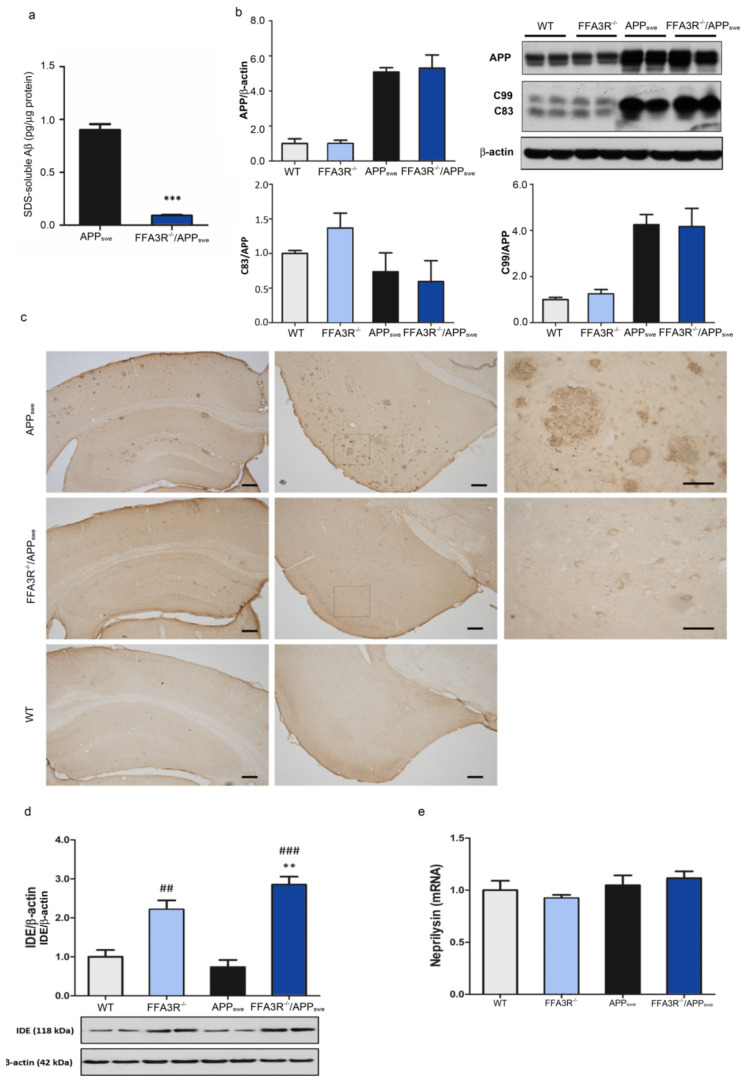
The FFA3R^−/−^/APP_swe_ mice show improved Aβ pathology and increased IDE levels. (**a**) Levels of soluble Aβ_42_ in the APP_swe_ and FFA3R^−/−^/APP_swe_ measured by ELISA. (**b**) Expression levels of full-length APP and APP carboxy-terminal fragments, C99 and C83, quantified by Western blot. (**c**) Immunohistochemistry for 6E10 in hippocampal and entorhinal cortex sections of WT, APP_swe_, and FFA3R^−/−^/APP_swe_ mice. Positive amyloid plaques are observed only in APP_swe_ mice. Magnification of 5× (Scale bar = 200 µm) in left and middle panels and 40× (Scale bar = 50 µm) in right panels. (**d**) Levels of IDE analyzed in WT, FFA3R^−/−^, APP_swe_, and FFA3R^−/−^/APP_swe_ mice by Western blot. We show a representative blot and the quantification of immunoreactive bands. Bars represent the ratio of IDE vs. β-actin. (**e**) Expression levels of neprilysin analyzed by RT-qPCR. Bars represent the enzyme expression normalized to that of the corresponding 36B4 internal control values. Results are expressed as mean ± SEM (n = 6) normalized to the ratio of WT mice. ## *p* < 0.01, ### *p* < 0.001 vs WT. ** *p* < 0.01, *** *p* < 0.001 vs. APP_swe_.

**Figure 5 ijms-23-03533-f005:**
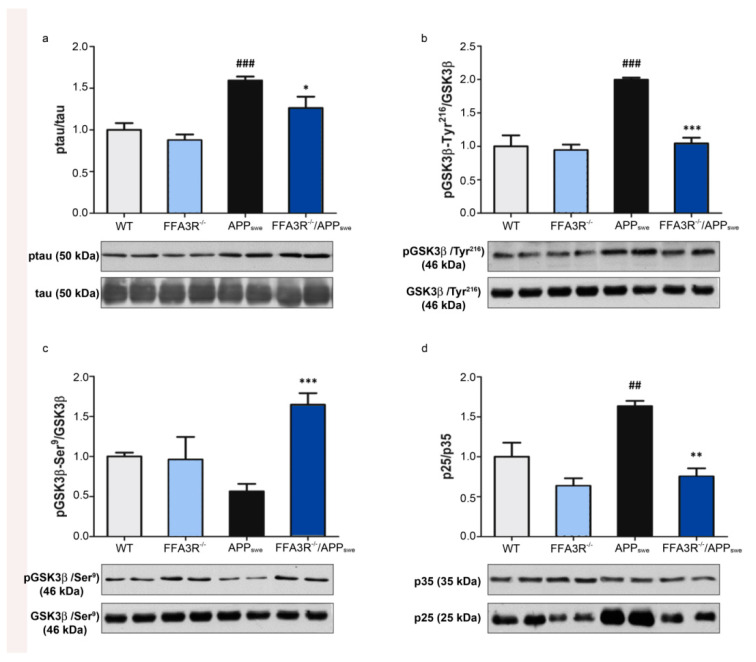
The FFA3R^−/−^/APP_swe_ mice show improved tau pathology. The levels of different proteins involved in tau pathology were analyzed in total hippocampal protein extracts of WT, FFA3R^−/−^, APP_swe_, and FFA3R^−/−^/APP_swe_ mice by Western blot. We show a representative blot and the quantification of immunoreactive bands for each protein normalized to the ratio of WT mice. The results are expressed as the mean ± SEM (n = 6). Bars represent (**a**) the ratio of ptau vs. tau, (**b**) the ratio of pGSK3β (Tyr^216^) vs. GSK3β, (**c**) the ratio of pGSK3β (Ser^9^) vs. GSK3β, and (**d**) the ratio of p25 vs. p35. ## *p* < 0.01 and ### *p* < 0.001 vs. WT. * *p* < 0.05, ** *p* < 0.01, *** *p* < 0.001 vs. APP_swe_.

**Figure 6 ijms-23-03533-f006:**
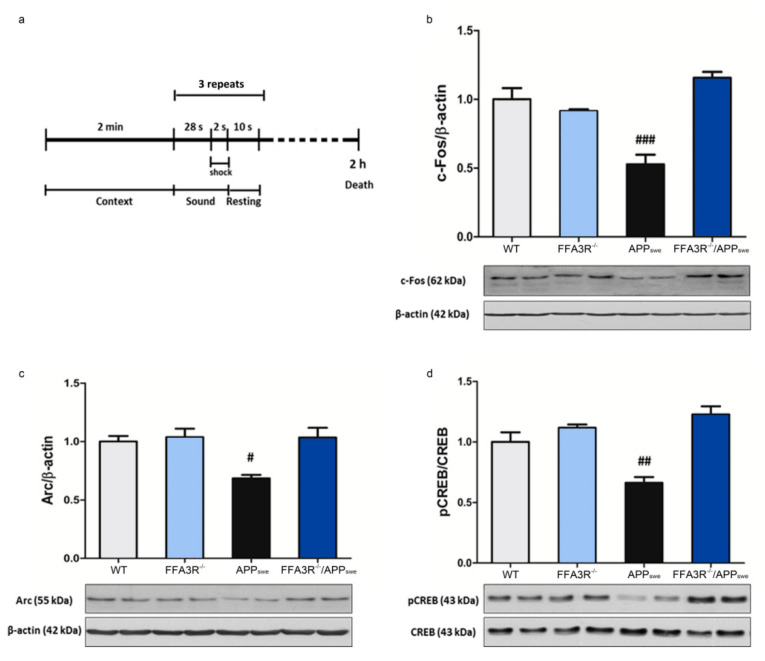
Induction of memory-associated genes in hippocampal mice samples. (**a**) Scheme showing the fear conditioning training. The levels of IEGs were analyzed in WT, FFA3R^−/−^, APP_swe_, and FFA3R^−/−^/APP_swe_ mice by Western blot. We show a representative blot and the quantification of immunoreactive bands for each protein. The results are expressed as the mean ± SEM (n = 6), and the bars represent the ratio of c-Fos vs. β-actin (**b**), the ratio of Arc vs. β-actin (**c**), and the ratio of pCREB vs. CREB (**d**) normalized to the ratio of WT mice. # *p* < 0.05, ## *p* < 0.01, ### *p* < 0.001 vs. WT.

**Figure 7 ijms-23-03533-f007:**
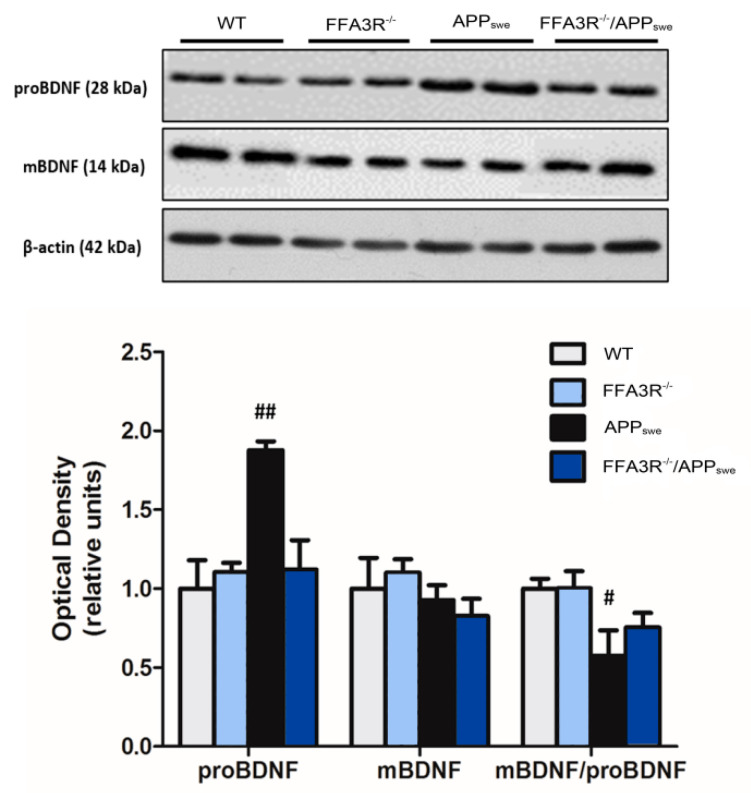
BDNF maturation is impaired in APP_swe_ mice and restored in FFA3R^−/−^/APP_swe_ mice. Levels of proBDNF and mBDNF were analyzed in total hippocampal extracts of WT, FFA3R^−/−^, APP_swe_, and FFA3R^−/−^/APP_swe_ mice by Western blot. We show a representative blot and the quantification of immunoreactive bands for each protein normalized to that of the corresponding β-actin internal control values. Results are expressed as mean ± SEM (n = 6) normalized to the ratio of WT mice. # *p* < 0.05, ## *p* < 0.01 vs. WT.

**Figure 8 ijms-23-03533-f008:**
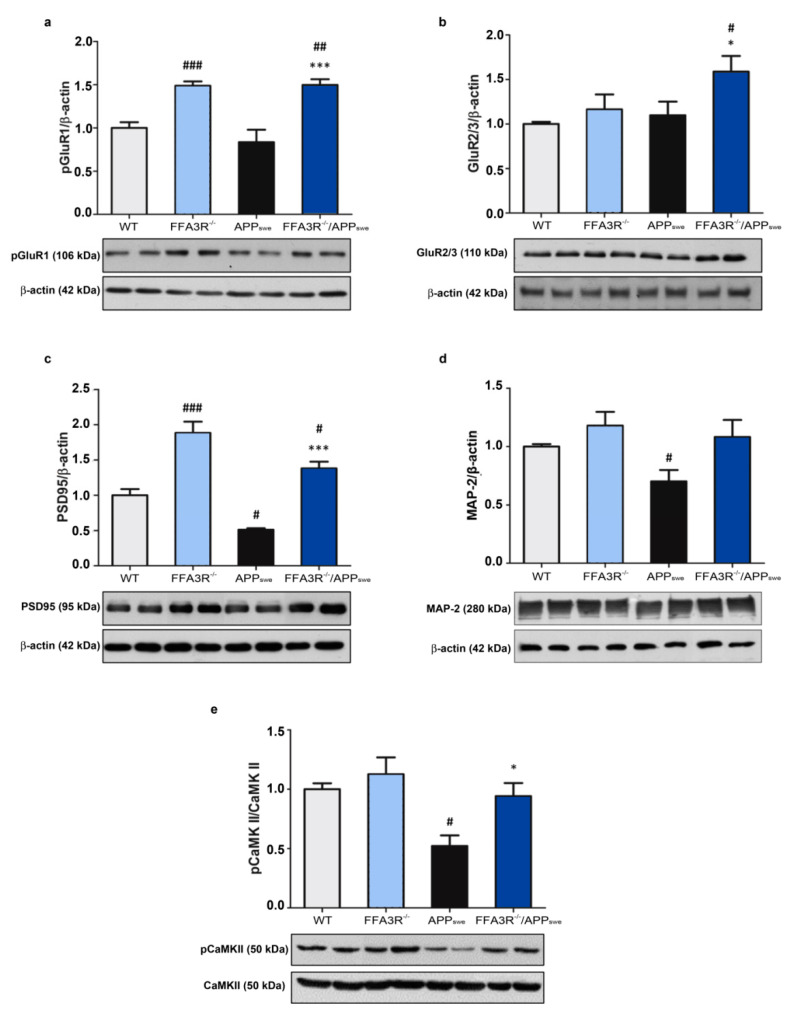
Synaptic plasticity markers are restored in the FFA3R^−/−^/APP_swe_ mice. The levels of synaptic markers in membrane cortex extracts of WT, FFA3R^−/−^, APP_swe_, and FFA3R^−/−^/APP_swe_ mice were analyzed by Western blot. We show a representative blot and the quantification of immunoreactive bands for each protein. The results are expressed as the mean ± SEM (n = 6) normalized to the ratio of WT mice. (**a**) AMPA subunit pGLuR1 vs β-actin. (**b**) AMPA subunit GluR2/3 vs. β-actin. (**c**) Cytoskeletal protein PSD95 vs. β-actin. (**d**) Microtubule-associated protein MAP-2 vs. β-actin. (**e**) Phosphorylated LTP mediator CaMK II vs. CaMK II. # *p* < 0.05, ## *p* < 0.01, ### *p* < 0.001 vs. WT. * *p* < 0.05, *** *p* < 0.001 vs. APP_swe_.

**Table 1 ijms-23-03533-t001:** Demographic and neuropathological characteristics of the subjects included in this study.

Braak Stage	Cases (n)	M/F	Age *, Year	PMD *, h
ND	6	4/2	35.5 (19–54)	3.2 (2.15–6)
Braak III–IV	4	2/2	83.5 (76–89)	2.5 (2–3.37)
Braak V	4	0/4	87 (77–92)	2.3
Braak VI	6	1/5	84.2 (79–89)	2.3 (1.25–3.3)

* Median and (range). ND = non-demented; F = female; M = male; PMD = postmortem delay; h = hours.

## Data Availability

The data that support the findings of this study are available from the corresponding author upon reasonable request.

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
