# Peer review of "Genetic Inactivation of Free Fatty Acid Receptor 3 Impedes Behavioral Deficits and Pathological Hallmarks in the APPswe Alzheimer’s Disease Mouse Model"

_ijms, 2022, doi:10.3390/ijms23073533_

Round 1
Reviewer 1 Report
Zamarbide et al. report about the expression and influence of FFAR3 in the hippocampus of Alzheimer’s disease patients and different transgenic mice (FFAR3-/-, APPswe and FFAR3-/-/APPswe). The study includes behavioral (Morris Water Maze), Abeta pathology, tau pathology and synaptic plasticity studies. The results show that the genetic inactivation of the Ffar3 gene in FFAR3-/-/APPswe mice do not lead to the characteristic memory impairment and do not develop the most important AD features (e.g. Abeta accumulation and tau hyperphosphorylation). Methods and data are very convincing. That’s why I recommend publishing this manuscript after minor revision.
- Delete “title” (first word) in the Headline
- Regarding the MWM: Have the authors looked at differences in gender (male vs. female)? If yes, could they provide the data?
- Line 222-224: Grammar! Please revise this sentence.
Author Response
The authors appreciate the reviewer's favorable comments and writing suggestions.
Following your instructions we have made the changes in the title and in the suggested sentence at the indicated lines.
We strongly agree with the comment related to the analysis of the MWM results taking into account the differences by gender. However, if we had done it this way, the number of animals per experimental group and sex would have been reduced to 4-6, depending on the experimental group. Obviously the statistical value of that analysis would have been very weak.
It is worth noting that the standard errors of the mean (SEM) are very narrow, which gives a good idea that the differences between the animals are small.
Reviewer 2 Report
The authors showed that the free fatty acid FFA3 receptor (FFA3R) occurs in the human hippocampus of Alzheimer disease (AD) patients and they demonstrated the FFA3R overexpression, even in the first stages of AD. On the basis of the results obtained from the mouse model with genetic deletion of the FFA3R they concluded that the brain FFA3R is involved in cognitive processes and its inactivation prevents AD-like cognitive decline and pathological hallmarks.
I think the authors presented an interesting work and the whole study seems to be well done technically. It is a well-written manuscript, and I only have minor comment:
In the legends to Figures: Fig 1, line 142 and Fig 2 line 159, please specify : ...Bars represent the receptor mRNA expression…… (instead of “the receptor expression…” only )
Author Response
The authors appreciate the reviewer's favorable comments and writing suggestions.
Following your instructions we have made the changes in the captions of figures 1 and 2
Reviewer 3 Report
Review of "Genetic inactivation of Free Fatty Acid Receptor 3 impedes 2 behavioral deficits and pathological hallmarks in the APPswe 3 Alzheimer’s disease mouse model"
This paper reads exceptionally well. The introduction of ism concise and to the point. The goals of the study are stated. The Results are very well done with labeling of the figure and details in the figure legends. The figures are clear and well-constructed for ease in understanding. This discussion highlights those main points and integrates well with the literature. It is nice the authors state how novel their findings are on the relationship within the CNS and AD as compared to a heavy past focus on the GI track. The Methods are detailed where others can follow in case, they wish to conduct similar experiments.
Overall, this manuscript read well and was very informative. This paper was a joy to read and to ponder over the implications for AD. I feel I number of investigators will be interested in reading this study.
I have no suggestive changes or additions to the manuscript.
Author Response
The authors appreciate the favorable comments of the reviewer.